# Exploring the Experiences and Current Support of Children and Young People with Selective Mutism Within Mainstream Secondary Schools

**DOI:** 10.3390/bs15070947

**Published:** 2025-07-14

**Authors:** Sophie Walker, Caroline Bond

**Affiliations:** Manchester Institute of Education, University of Manchester, Manchester M13 9PL, UK; caroline.bond@manchester.ac.uk

**Keywords:** selective mutism, children and young people’s experiences, secondary school, support, progress

## Abstract

Few studies have explored the views of children and young people (CYP) with selective mutism (SM), and even less is understood regarding their experiences in relation to the support that they receive within school. Across three case studies, direct interviews with CYP with SM attending mainstream secondary school were conducted non-verbally, aiming to explore their current experiences of school and support. Subsequent interviews were conducted with the CYP’s key stakeholders, including parents/carers, school staff, and professionals with ongoing involvement. These interviews aimed to build on information shared by the CYP. Analysis highlighted the importance of individual experiences and support, relationships with peers and trusted adults, collaboration, communication across the setting, and importantly, a secure understanding of SM across the school setting. Clear implications for school professionals emerged. Future research should continue to work toward the exploration and development of knowledge and understanding of SM and gather the experiences of a wider range of CYP and families.

## 1. Introduction

Over recent years, there has been increasing focus on the role schools play in supporting CYP’s mental health and access to support. The Department for Education (DfE) co-produced a green paper with the Department of Health and Social Care (DHSC) in 2017, highlighting the enhanced role schools occupy across the domains of identification, planning, and support regarding CYP’s emotional well-being and mental health ([12]). Wider legislation and policy have also acknowledged that CYP with special educational needs (SEN) should have the opportunity to share their views to inform this support ([9]; [41]). Despite this, there continues to be a lack of representation of the views of CYP with SM within the sparse literature ([44]), and little understanding about the school context within which these CYP’s difficulties occur ([44]). This lack of information appears to be impacting these CYP’s access to the appropriate identification, planning, and support that they need and are entitled to within school.

This paper synthesizes and presents pertinent literature that explains how SM is currently understood within the research field. The prevalence and current types of support recommended for SM will be highlighted, as well as the importance of considering ways to gather this group’s views. Finally, the present study addresses the ‘gaps’ identified in the literature review.

### 1.1. Selective Mutism

SM is an anxiety condition where a person is unable to speak in certain social situations where there is an expectation to do so, such as school, despite being able to speak fluently in other settings, such as home ([1]). Research indicating anxiety is a central part of SM ([35]) has supported the change in terminology from ‘elective mutism’ to SM ([1]). This linguistic reframing demonstrates a shift in the understanding of SM from an act of will, to a lack of ability to speak in certain situations ([26]). The [34] ([34]) has furthered understanding by describing SM as a ‘speech phobia’, where the expectation to speak elicits high levels of anxiety and causes a physiological ‘freeze’ response, leaving a person unable to speak ([19]). The condition typically begins in early preschool years, ages 2–4 ([25]), and for a diagnosis, the absence of speech must be present for at least 1 month and not be attributable to a lack of knowledge of, or comfort with, the spoken language required in the social situation (DSM-5; [1]).

Research on the prevalence of SM is limited, with no agreed incidence at present ([13]); however, NHS data indicated that 1 in 140 primary age children have an SM profile ([25]), reducing to 1 in 550 for children up to 15 years ([37]) and approximately 1 in 2400 (0.04%) in adults ([40]). However, it is important to consider that older school students with a SM profile may have never received an appropriate diagnosis and/or intervention, and therefore have experienced negative reinforcement during school, leading to ingrained behaviour patterns and maladaptive coping mechanisms ([36]). Given current prevalence rates, most secondary educational settings are likely to have at least one child with SM, and potentially many more without a formal diagnosis ([20]).

### 1.2. Presentation

Presentations of SM vary between individuals and are situational. Current literature highlights a distinction between high-profile and low-profile SM. The Selective Mutism Information and Research Association (SMIRA) proposes that ‘high-profile SM’ is characterized by CYP who are unable to speak at all to certain people, making them more observable due to their contrasts in speaking patterns. Those with ‘low-profile SM’ may occasionally speak when prompted, with others often regarding them as shy, quiet, or rude, without recognizing that speech provokes the same intense anxiety as those CYP with ‘high-profile’ SM. It is also important to acknowledge that CYP can fluctuate between both presentations depending on a range of factors such as support, communication load, and the wider environment. A lack of understanding of this can often result in a lack of appropriate support. This is a concern as CYP with ‘low-profile’ SM are at an increased risk of presenting as ‘high-profile’ if they are not supported effectively ([19]).

Although not highlighted within the diagnostic criteria, anxiety, including social anxiety disorder (SAD), is a commonly co-occurring difficulty for children with SM ([23]). Alongside this, research demonstrates evidence of a clear relationship between SM and Autism ([24]), although the [1]’s ([1]) and International Statistical Classification of Diseases and Related Health Problems (ICD-11) definitions are somewhat ambiguous and could be interpreted as implying that ASD and SM are mutually exclusive ([22]). Despite this, the overlapping nature of these conditions is reflected in many children often receiving co-occurring diagnoses of ASD and/or SAD alongside SM within practice.

### 1.3. Support and Intervention

Across the limited literature, there is a range of approaches to supporting CYP with SM ([31]). Within the UK, support often begins with a graded exposure approach, supported by strategies within the school and home ([19]). [19] ([19]) advocate for a holistic approach when formulating comprehensive interventions to support SM. A review by [17] ([17]) identified that the most common method of intervention approach involves collaboration between clinicians, parents, and teachers. Across the research field, early intervention and identification of SM have led to a greater chance of successful treatment ([3]). However, [4] ([4]) highlighted the lack of available support for SM children and adolescents within the existing literature. This is a concern as longer-term SM has been found to share a range of co-morbidities such as social anxiety disorder, depression, and suicidal ideations ([8]). [5] ([5]) highlighted that a lack of knowledge and understanding of SM in adolescence can lead to interventions focusing on elicitation of speech rather than a more holistic understanding of the young person. This resulted in the call for research that explores potential environmental, intrinsic, and psychological factors that may lead to, and perpetuate, SM before more formalized treatment approaches are made available.

### 1.4. Views of CYP with SM

Earlier research from [10] ([10]) highlighted the lack of the child’s lived and living experience within the SM literature. A review of 140 articles on SM by [33] ([33]) found that 112 of the papers related to individual cases, but only two of those included the direct views of CYP with SM ([27]; [29]). To access the young people’s views, [29] ([29]) interviewed three CYP with SM using the Raven’s Controlled Projection for Children (RCPC), which enabled children to communicate in writing about their perspectives, wishes, and experiences, removing the pressure to contribute verbally. Building on these methods, [33] ([33]) used questionnaires completed at home with around 30 CYP aged 10–18. A separate parental questionnaire, both designed to elicit strategies for communication, help and hindrances to recovery, and their opinions and feelings about SM, was also shared. There is evidence to show that 80% of YP felt that SM had affected them in school and resulted in missed opportunities ([33]). Furthermore, an unpublished thesis from [43] ([43]) used a non-verbal adaptation of a technique informed by personal construct psychology within online interviews to explore CYP with SM’s experiences of school. Findings highlighted that wider external factors in school, e.g., the role of staff and peers, school layout and school ethos, and internal factors concerning the participants, e.g., identity, confidence, and trusted adults, both contributed to the maintenance of SM or progress of CYP. Despite these studies, there continues to be a lack of research that explores the relationship between school experience, support for this cohort, and the use of triangulation from key stakeholders.

The present study explored the perspectives of secondary school-aged CYP with SM, triangulated their experiences through the insights provided by the CYP themselves and key stakeholders, and provided a clearer understanding of the underlying principles and mechanisms that underpin effective support for this population of CYP. This integration of different perspectives aims to create a more holistic understanding of this process in context. The present study reports authentic data that reflect the views of CYP with SM in relation to their current experiences and support within school.

### 1.5. Research Questions

How do secondary school-aged young people with SM experience school?Can we define and identify effective support for young people with selective mutism within secondary school?

## 2. Materials and Methods

### 2.1. Epistemological Position

The ontological and epistemological positioning of the researcher has influenced the design of this research. The study design was influenced by the research team’s critical realist approach, which combines a realist ontology with a constructivist epistemology ([15]). Critical realism emphasizes the importance of understanding that knowledge can be interpreted differently by different individuals within and across contexts ([32]). By combining these perspectives within a case study approach, the researcher was able to consider both the objective structures that influence and shape reality, and the subjective interpretations and experiences that individuals perceive about that reality.

### 2.2. Design

This study adopted an exploratory multiple-case study design ([46]), aiming to explore CYP’s school experiences and current support within school. The ‘cases’ were three YP within three different schools. The emphasis of this research was to capture their lived and living educational experiences and better understand how they, and adults who know them well, experience support for the young person within that context. Interviews were chosen to collect data from key stakeholders, including parents, teachers, and professionals. [46] ([46]) highlighted that exploratory case studies are needed to help researchers understand complex phenomena within real-life settings, enabling an in-depth analysis and deeper understanding of a topic. Given the lack of research investigating the lived and living educational experiences of CYP with SM, this research design was the most appropriate starting point for developing a deeper understanding and expanding the knowledge base.

### 2.3. Participants and Sampling

Participants were of secondary school age, had no ongoing safeguarding concerns, attended mainstream provisions, and could have several co-morbid diagnoses. Participants were recruited across a range of research and professional networks, and parental consent was obtained before work commenced. The work was organised through independent gatekeepers within each of the three participating schools, who were all special educational needs coordinators (SENDCo). The participating schools were from socioeconomically diverse communities in the Northwest of England. The sample consisted of two females and one male participants. For each of the participants, a range of key stakeholders was recruited (see Table 1 below for details).

### 2.4. Data Collection Methods

Phase 1: Gathering informed assent and building rapport

The first phase of work focused on building rapport and gaining informed assent from each participant. Assent was gathered non-verbally through a child-friendly assent form and information sheet, completed with support from a trusted adult. Any questions could be asked and answered non-verbally within the session, through parents after the session, or via the young person’s trusted adult at school. This session was planned to ensure that the expectation to speak was removed, aiming to reduce anxiety and ultimately make this process as accessible as possible for CYP with SM ([11]; [19]).

Phase 2: Gathering young people’s experiences

The focus of this phase was to obtain a more detailed understanding of the CYP’s secondary school experiences. The CYP were given the opportunity to choose a preferred method of communication through a visual prompt. Methods offered included written responses, drawing, talking mats, and a questionnaire developed by the researcher. The researcher, as a practitioner psychologist, utilized their skills via attunement and previous knowledge of SM to continuously adapt and respond to the CYP within the session, ensuring their emotional needs were met throughout. Research has highlighted that educational psychologists are well placed to obtain young people’s views that reflect a true representation of their experiences ([39]). After the session, the researcher’s observations were recorded

Phase 3: Gathering information from key stakeholders

Following pupil interviews, a range of key stakeholders who knew the CYP well were interviewed. These included parents, form tutors, key workers, and professionals with ongoing involvement. These interviews aimed to gather information regarding their understanding of selective mutism, views on current support in school, explore areas for further development, and gather the adults’ own perceptions of the young people’s experiences. The interviews were used as supplementary data to a greater or lesser extent, depending on the CYP’s ability to communicate their views to the researcher directly within the direct work. Contextual conversations about each of the settings were also held with the independent gatekeeper, SENDCo, for each case study to better understand the SEN provision and school cohort/characteristics, which are presented within the findings.

### 2.5. Data Analysis

Reflexive thematic analysis (RTA) was utilized to analyze the data collected from semi-structured interviews with key stakeholders using the six-step method outlined by [7] ([7]). The 15-point checklist for ‘good thematic analysis’ and the six-phase guidelines for conducting thematic analysis as described by [6] ([6]) were followed. As reflexive thematic analysis is a process of interpretation of meaning rather than a discovery of an accurate truth or reality, inter-rater reliability measures were not used within this study. Instead, a collaborative and reflexive approach was taken between the authors to find meaning from the analysis that was completed ([7]).

When interpreting the CYP’s interviews with the researcher, given their contrasting presentation and contributions to the sessions, a double hermeneutic approach was adopted ([16]). This helped to make sense of and capture the wider, holistic interaction between the researcher and the CYP, and was completed with the intention of drawing out practical implications to support practitioners’ knowledge and understanding of how best to support CYP with SM within communicative exchanges with professionals. To analyze the CYP’s contributions (talking mats and written questionnaire responses), guidelines for conducting thematic analysis with written and pictorial data, as described by [7] ([7]), were used with all collected data The analysis was semantic and inductive, allowing themes to emerge from participants’ contributions; however, there were also wider latent aspects of the analysis that captured the deeper underlying assumptions from the data that are discussed in the analysis and cross-case analysis. After completing each case study analysis, a cross-case analysis was completed to capture any common themes across the young people’s experiences, and importantly, to draw out themes across the support that may benefit all CYP with SM.

## 3. Results

Findings are presented on a case-by-case basis, followed by a cross-analysis; italics represent direct quotes either shared verbally or written.

### 3.1. ‘Olivia’

#### 3.1.1. Context

Olivia was in KS3 and had co-occurring diagnoses of low-profile SM, autism spectrum condition, and social anxiety disorder. Contextual conversations with the SENDCo highlighted embedded communicative approaches within the school, including the consistent use of pupil passports for all pupils with SEN, supported by quality first teaching. Staff had received whole-school training on SM, and collaboration between all stakeholders, including Olivia, was embedded within practice.

#### 3.1.2. Interview Process

Before working with Olivia, important considerations were made regarding the room choice, seating, and offering trusted adult support. Starting with a familiar task, talking mats helped reduce anxiety and provide predictability. The researcher used a ‘commentary’ style communicative approach to reduce the expectation to speak, and to provide Olivia time to regulate herself. With this reduced communication load, Olivia began to speak spontaneously during the activities. Verbal and visual prompts helped scaffold Olivia’s responses, enabling further detail and clarity to be established. The researcher was attuned to Olivia’s non-verbal communication, ending an activity when needed. Simplified questioning and non-verbal communicative methods were key in creating predictability and enhancing interactions.

Analysis of Olivia’s talking mat and questionnaire responses (written and verbal) identified five broad themes. The following reported findings relate to the themes depicted in Figure 1.

#### 3.1.3. Environment

Olivia identified challenges within the mainstream school environment, which led to feelings of distress. These included her experience of the sensory environment, such as wearing school uniform, and the noise levels in school, explaining ‘in form it is loud, but I like it quiet’. Olivia highlighted opportunities to access quiet spaces in school as a supportive measure. Similarly, the environment Olivia was in impacted her ability to communicate with others, as she was better able to communicate verbally when the situational demands were reduced, particularly within a conversation with a familiar peer or staff member when speaking one to one.

#### 3.1.4. Subjects

Olivia appeared to be at her best in school when engaging with her preferred subjects: Art and English. She found performative subjects in school, such as sports or Spanish, challenging. Olivia explained, ‘I don’t like sports, like rugby and dance because everyone is looking at you like in netball too’, highlighting that her feelings and experiences of being watched made her engagement with these areas of the curriculum difficult.

#### 3.1.5. Relationships

Relationships with staff and peers were an important area of support for Olivia in school. She received support from her peers and could communicate with them confidently when friendships were established. Olivia also found it easier to talk to familiar adults, stating, ‘it’s easier when I know them’. However, if any communicative partners placed increased demands on Olivia within conversation, this could be difficult for her to manage, leading to feelings of dysregulation and burnout (tiredness), reducing Olivia’s capacity for challenge and willingness to attend school. She explained when interacting with peers, ‘sometimes I just think be quiet and stop talking about geese [private joke]’, and when adults use humor, ‘it depends if they’re actually funny?!’

#### 3.1.6. Communication and Interaction

Olivia expressed some difficulties with the social aspects of communication. For example, she found it difficult to understand body language, initiate conversation, and end conversations, explaining ‘it’s thinking of the right thing to say [to end an interaction], like sorry I need to go for a shower’. Olivia found it difficult to explain how she felt within these situations, stating ‘if I can’t speak I forget about it afterwards’ and ‘I don’t remember’ or ‘I don’t know’. It was clear these situations caused Olivia to feel increasingly anxious.

#### 3.1.7. Reasonable Adjustments

Olivia highlighted that she felt her current support in school was helpful. In particular, she highlighted that being able to share how she was feeling with trusted people was important, stating ‘a familiar face can help’, and ‘more time with my key worker’ would support her further. Some reasonable adjustments, such as opportunities for time on her own and access to a quiet environment, were also helpful. However, Olivia acknowledged that she was not able to access some of the support strategies in place, for example, her time-out pass, seemingly because of her difficulties initiating interactions.

#### 3.1.8. Parent and Staff Perspectives

The following findings will relate to the themes depicted in Figure 2 below.

Interviews from Olivia’s parents, key worker, and the assistant psychologist with ongoing involvement with Olivia described an evidence-based package of support that worked to adapt the sensory environment, support Olivia’s emotional well-being, and provide alternative means of communication where possible. Strategies were informed by all key stakeholders, including Olivia’s views.

Collaboration between home, school, and external agencies was frequent and supported through the development of trusted reciprocal relationships that had developed over time, and were maintained through ongoing communication. The approach taken by all to develop this collaborative relationship was important; parents were proactive and informed, and Olivia’s key worker explained, ‘I think we can also be the middle person between the teacher and student’, acting as a mediator to facilitate communication between stakeholders.

To collaborate effectively, all key stakeholders had worked to build their knowledge and understanding of selective mutism. It was clear that Olivia’s parents had to self-educate not only about her diagnoses, but also to understand educational systems and the appropriate provision to support SM in school, stating: ‘self-education is really important’. Similarly, staff mostly had a good understanding of SM and how this impacted Olivia’s presentation, as her father stated: ‘They understand. They have put things in place, that keep some of [child’s] anxiety levels at, at, a manageable level’, which had been embedded through whole school training. All interviewees acknowledge the vital role of training when working with SM CYP.

Due to effective collaboration, support for Olivia was informed, meaning that school staff anticipated areas of difficulty for her, and scaffolded her engagement through a range of reasonable adjustments and attuned check-ins that were accessible and met her needs. Her key worker explained, ‘if she’s had a freeze, I can’t straight away keep asking her. I’ll say, ‘ok I’ll give you some time then I’ll come back to you’. Olivia’s support included additional support for transition, unexpected change, and performative subjects—as her parents raised their ongoing concerns about the demands such changes and situations placed upon Olivia. Olivia sought support from her peers to advocate for her needs and wants. School enabled her to access that support where necessary through class groupings and seating plans. Again, this practice reflected a person-centered approach and ethos within the setting when supporting children with additional needs.

### 3.2. ‘Catherine’

#### 3.2.1. Context

Catherine was in KS3 and had co-occurring diagnoses of SM, autism spectrum condition, and ongoing medical needs. Contextual conversations with the SENDCo highlighted high levels of SEN within the setting. Children with SEN were allocated key workers to support both their academic success and meet their emotional needs. Children had a pupil passport that staff within the wider school could access, which outlined their support needs and commitments. The SENDCo acknowledged the importance of consistency, collaboration between stakeholders, and building a safe space within the SEN provision for young people to access when needed.

#### 3.2.2. Interview Process

The researcher considered seating arrangements and ensured minimal disruptions before working with Catherine. Catherine used non-verbal communication such as nodding, gestures, and written responses throughout the session and wrote down more complex thoughts or responses when needed. Mutual interests in animals facilitated rapport building and led to an instance of spontaneous speech from Catherine. Catherine was supported to access the interview by allowing her additional time to process information, the use of non-verbal communicative approaches, and a reduction in communication load through a ‘commentary style’ approach.

Analysis of Catherine’s written questionnaire and talking mat responses identified four broad themes.The reported findings relate to the themes depicted in Figure 3.

#### 3.2.3. Relationships

Relationships with staff and peers were an important area of support for Catherine in school. She explained that her ‘caring friends’ provided her with support in school, and she found interacting with them significantly easier than with adults. Catherine noted that she hoped to be ‘speaking to them more and making sure they’re ok’ moving forward. Alongside peer support, Catherine benefited from relationships with trusted adults at home and school. It appeared that this support helped her to regulate within school, writing, ‘it is very tricky to talk but if you have trusted adults, you feel calmer’.

#### 3.2.4. Difficulty with Communication Skills

Catherine shared her difficulties with a range of skills needed for communication, including thinking, sharing complex information, problem solving, and remembering, within the talking mats activity. Additionally, she highlighted struggles with navigating the social aspects of communication, such as knowing when to begin or end a conversation. It may be that these difficulties were further contributing to her identified difficulties related to attention and focus, as she was anxious and dysregulated within school. Catherine hoped ‘having something to [help] focus’ would support her further in the future.

#### 3.2.5. Communicative Demand

Consideration of how to reduce communicative demand was important to support Catherine. Meeting new and unfamiliar people in new and unpredictable environments was difficult for her and could make her feel ‘pressured with them’. If people/staff implied an expectation for Catherine to speak, this could lead to feelings of distress for her, as she shared, ‘others [teachers] say I have to speak, which makes me uncomfortable’. Catherine reported non-verbal means of communication within school had enabled her to demonstrate how she was feeling more freely, as she wrote, ‘you can express how you want in dance without using words’. She also indicated that visual aids, writing, and not having to speak within lessons were helpful.

#### 3.2.6. Consistent Support

Catherine appeared to be largely happy with her school support, and she hopes that in the future she will feel ‘more comfortable’ when interacting with staff. However, for this support to be effective, her views implied the need for increased consistency across the staff supporting her. She shared it was not helpful when ‘teachers not listening to the plan of me sitting next to my friend’.

#### 3.2.7. Parent and Staff Perspectives

The following findings will relate to the themes depicted in Figure 4 below.

Parent and form tutor interviews highlighted the important role of **communication and collaboration** to support Catherine. While parents and the SEN team shared important information to support her, Catherine’s form tutor acknowledged she often only collaborated with home ‘as and when I need to, sometimes’. SEN staff tried to use school communicative systems to relay information from parents to the appropriate staff working with Catherine. Her form tutor acknowledged the need to gather Catherine’s views to inform support, recognizing this is a current gap in her understanding of Catherine and her presentation. Catherine’s mother valued external professional support, but had faced challenges in accessing these services post-diagnosis.

Discussion on the significance of **relationships** within school for Catherine highlighted her progress within this area, as her mother pointed out ‘she’s definitely found a voice more, and is telling people[friends] no if she’s not happy with the situation’, highlighting her increased confidence in advocating for herself with her friends. Her form tutor further highlighted that Catherine preferred to communicate more complex responses and ideas through her closest friends. Regarding her relationships with adults, Catherine had developed some trusted relationships after becoming familiar with them through regular check-ins.

Despite efforts to build relationships and collaborate among key stakeholders, there remained a lack of **knowledge** of selective mutism within school, which was impacting Catherine’s progress. Catherine’s mother highlighted misunderstanding, explaining ‘the only barrier has been teachers and teachers who have said to me, what do you means she’s autistic and a selective speaker, no she’s not?’ Catherine’s mother had worked to develop her own knowledge of SM, drawing on resources from her own job in a primary school. The importance of potential future training was acknowledged by all.

Insights from parents and staff identified school elements that posed a challenge for Catherine, including the need to **reduce demands** within the sensory environment, situational demands, and the option to communicate non-verbally, especially when experiencing being overwhelmed. However, when coupled with a misunderstanding of SM, it appeared that there were still some unhelpful and challenging communicative demands being placed on Catherine within school. Her mother explained, ‘the only time she struggles, and she will say, is when she feels forced to speak.’ This misunderstanding was further acknowledged within the staff’s **approach to supporting Catherine**. Despite some things going well, it was not always clear what had contributed to Catherine’s progress, and how to best develop her skills further. Her form tutor wondered whether further opportunities to ‘practice’ skills within a safe space may be helpful for Catherine.

### 3.3. ‘Michael’

#### 3.3.1. Context

Michael was in KS3 and had selective mutism. Contextual conversations with the SENDCo highlighted that the school had difficulties with staff recruitment and retention, alongside increased levels of SEN, with more children presenting with communication difficulties. A lack of access to quality training and external support from professionals, such as speech and language therapy, was also leading to some staff not always having a clear understanding of SM and its etiology. Children with EHCPs accessed support in class from an allocated key worker, and had a personalized one-page profile outlining their needs and support. The SENDCo acknowledged the importance of peer support for Michael in school, and the need for reasonable adjustments.

#### 3.3.2. Interview Process

Before working with Michael, the researcher introduced herself to him when interviewing his mother online. Michael shared his interest in music with the researcher by bringing his instruments within sight of the camera. The researcher prepared Michael for their meeting at school through reminders via his parent and key worker. The open workspace allocated for the interview within the school was not helpful for Michael’s access to the work, as he presented as anxious and had a frozen smile. To reduce demands further, the researcher clarified the task, ensured clear expectations, used humor, and a ‘commentary’ style approach to communication. Initially, Michael struggled to access the task, but with adapted approaches and reduced demands, he was able to independently sort cards and build confidence throughout. Despite Michael not being able to clearly advocate for his wants and ideas within the session, he did benefit from the adaptations, and his key worker mirrored the researcher’s approach during the interview, reducing her demands and enabling Michael to engage more confidently. The session ended positively with a game, during which Michael smiled.

#### 3.3.3. Themes

Analysis of Michael’s talking mat and limited questionnaire responses identified three broad themes.

The reported findings relate to the themes depicted in Figure 5.

#### 3.3.4. Building and Developing Relationships

Michael currently finds talking to friends in school significantly easier than interacting with adults. When conversing with friends, Michael can communicate freely, even when teachers are nearby. Interacting with unfamiliar individuals proves challenging for Michael, but as he grows more familiar with people, interactions with them can become easier. Michael was able to confidently indicate his interests using the talking mats, and it was clear that when supported by family and friends, he can access a range of hobbies and community activities such as the cinema, holidays, or eating out.

#### 3.3.5. Strengths in Skills for Communication

Within Michael’s response to the talking mats, he indicated his confidence with a range of skills needed for communication. Specifically, he expressed his perceived proficiency in understanding language, his thinking skills, his listening skills, and use of humor as a communication tool.

#### 3.3.6. The Need for Further Support

Michael’s responses clearly demonstrated how difficult he currently finds interacting with school staff within school. Discussion of this topic led to him becoming dysregulated. It also appeared that other elements of school posed additional challenges for Michael, notably, his ability to remember information, engage in new things, and participate in religious studies lessons.

##### Parent and Staff Perspectives

The following findings will relate to the themes depicted in the Figure 6 below.

Interviews with Michael’s parent and key worker highlighted a lack of understanding of selective mutism within school, with his key worker commenting on a small group intervention: ‘I don’t know how much it will achieve with him purely because it is his choice not to speak’. Michael’s mother highlighted the importance of **knowledge** in being able to appropriately support SM, sharing her self-education after Michael’s diagnosis and subsequent role as an advocate for understanding of the condition with school professionals and wider family, stating ‘once we started to, like, explain it, they were very supportive of him’. Interviewees both acknowledged that school staff would benefit from further training to help develop knowledge regarding SM, and provide appropriate support.

Due to this lack of understanding within school, there continued to be a range of **demands** placed on Michael that appeared difficult for him to navigate. His key worker noted difficulties in the sensory environment, and in using non-verbal communication methods such as visual aids or cards, stating ‘you don’t get any kind of acknowledgement back that he’s understood’. Michael continued to freeze during check-ins and was not able to access the adult support that the school was providing. Contrastingly, Michael’s peer **relationships** appeared to be vital to his access to school. His key worker explained ‘he had [peer] sat next to him, and that was it then, [he] perked up, just chatting away’. Peers supported Michael to advocate for his needs and wants within school.

**Communication and collaboration** were beginning to support Michael within school, for example, an enhanced transition helped him settle in well. School staff often contacted Michael’s mother for guidance on supporting him, and shared positive progress with her. She explained that Michael’s form tutor ‘sent me an e-mail on a Sunday night in his own time to say, ‘I’ve got a music lesson with [child] next week I want some pointers on how to approach it’. However, due to the school staff’s lack of knowledge and experience in supporting SM, further collaboration was needed to develop appropriate and informed support for Michael.

Interviewees both shared concerns about the **impact of SM** on Michael. His key worker expressed her worries about his future access to the curriculum and his asking for help if he was not able to develop his verbal communication skills. His mother explained Michael can engage in some avoidance strategies on school mornings, sharing, ‘it’s hard to get him moving in the mornings with the anxiety though, he struggles to sleep at night’. This highlighted clear elements of the wider impact of Michael’s difficulties on his well-being.

### 3.4. Cross-Case Analysis

A cross-case analysis highlighted several key patterns and differences across the three cases. A consistent finding was the vital role of peers as a source of support within the school environment for all young people. However, nuances emerged in how each individual navigated peer relationships and the social demands they entailed. Olivia and Catherine appear to face some additional challenges in navigating these relationships and the demands they present, particularly due to the heightened communicative demands and social anxiety associated with their co-occurring autism diagnoses. In contrast, Michael appeared more confident and fluent in peer interactions, which may reflect differences in the presentation or impact of selective mutism when co-occurring with Autism.

All three young people had an allocated key worker and benefited from time to become familiar with them, leading to the development of a more trusted relationship. Nevertheless, differences in the consistency and effectiveness of these relationships were evident. For example, Catherine’s support was undermined at times by inconsistency in staff’s adherence to her individual support plans, whilst Michael’s key worker acknowledged systemic challenges such as limited staff training and high workload.

Frustratingly, in both Catherine’s and Michael’s cases, due to a range of factors, including lack of funding, staffing resources, and the reduced capacity of local authority mental health services, participating schools were unable to make onward referrals for further psychological support for Catherine and Michael. This contrasted with Olivia’s more embedded and collaborative support system, reflecting variable access to resources/services directly impacting support quality and outcomes. Feelings of frustration emerged within the researcher’s personal research diary related to these extraneous circumstances.

Interestingly, Olivia and Catherine appeared to struggle more with their communication skills in comparison to Michael, suggesting possible implications for the relationship between SM and ASD. All three case studies demonstrated the impact of various demands within school, such as the physical environment and the expectation to speak, leading to anxiety and a ‘freeze’ response. However, Olivia showed particular sensitivity to performative subjects (e.g., sports, dance, languages), which were experienced as more anxiety-provoking than for Michael and Catherine, possibly due to her greater social anxiety and sensory sensitivities.

Stakeholder interviews highlighted the importance of communication and collaboration, trusted relationships, and understanding of SM. Differences in how well these factors are embedded into current support packages have impacted how informed, appropriate, and effective they appeared to be.

Overall, the comparative analysis reveals that while peer support and key worker relationships are universally important, individual differences in communication skills, co-occurring conditions, school resource availability, and staff training critically shape the effectiveness of support for young people with selective mutism. These divergences underscore the necessity of tailored, well-resourced, and collaborative approaches to meet diverse needs within educational settings.

## 4. Discussion

### 4.1. Key Findings and Implications for Practice

This study makes an original contribution to our understanding of CYP with SM’s school experience and support. The support these CYP receive plays an important role in shaping their school experience more generally. Findings suggest it is imperative that support is informed by a thorough understanding of SM to effectively alleviate some of the challenges that these CYP experience, and to promote opportunities for progress. Effective communication and relationships are needed between all key stakeholders, including home, school, external professionals, and the CYP themselves, to ensure the successful implementation of strategies.

### 4.2. Knowledge and Understanding

Previous research from [45] ([45]) acknowledged that school staff have often felt ill-equipped from their CPD and initial teacher training to be able to recognize signs of SM, highlighting that further training is needed to fill this gap in knowledge. Similarly, the findings of the present study echo a continued misunderstanding of SM as a ‘choice’, based on questions raised around the use of the term ‘selective’. Importantly, it appears that a misunderstanding of SM is impacting CYP’s access to appropriate and informed support within school. As highlighted by [5] ([5]), a lack of knowledge and understanding of SM in adolescence often leads to interventions focused on elicitation of speech, rather than working to develop a more holistic understanding of the young person. [5] ([5]), as well as other researchers such as [21] ([21]), has called for further exploration of these potential environmental, intrinsic, and psychological factors that may be leading to and perpetuating SM. Findings from this study position a lack of understanding of SM as one of the clear maintaining factors that continues to perpetuate SM for these CYP. [19] ([19]) explain maintaining factors to be ‘behaviours and events that strengthen and maintain the fear, to the point where no amount of reassurance or logic can dispel it’. For example, within the findings, some staff continue to apply pressure to speak within school. This pressure to speak then acts as a maintaining factor, ‘strengthening their [young people’s] conviction that talking is too difficult; each time this happens their fear of talking increases’ ([19]). Therefore, it is key that knowledge and understanding are developed to eliminate these maintaining factors, as if left unsupported, SM is associated with increased rates of social anxiety disorder, depression, and suicidal ideations ([8]).

### 4.3. Relationships

Results showed that participants regarded relationships as a key part of the school experience and support package. As acknowledged within the literature, CYP with SM often are able to speak to close friends, and these friends play a vital role in supporting and advocating for the CYP’s needs and wants within school ([18], [19]). Findings highlighted the importance of school staff working to facilitate these relationships by considering seating plans alongside peers, and providing opportunities to connect during the school day. Although limited, previous research acknowledges the importance of peer inclusion for CYP with SM, and settings working to facilitate the development of these relationships by involving them in activities alongside peers can act as a facilitator in their progress ([28]).

Earlier research indicates that an understanding of SM as a ‘speech phobia’ is required (i.e., where a physiological freeze response can occur when communicating with adults within the school setting ([19])). Findings indicated that building rapport over shared interests, providing check-ins, and trust building over time led to established relationships with staff. It was clear that CYP benefit from support from adults in school. Where these relationships were most successful, it appears that staff had a good understanding of how to reduce demands for the CYP. These adaptations included consideration of the potential sensory demands, current situational demands, or reducing their own communication load when interacting with the YP. Research from [19] ([19]) highlighted the important role of working to reduce our communicative demand when interacting with CYP with SM. Lower-load activities may involve the use of commentary-style language from the adult, and rote language or factual single-word responses needed from the CYP. High-load communication activities involve asking for ideas/opinions, when a CYP is unsure of the answer, or unplanned communication. Removing this level of demand and often providing non-verbal methods of communication reduced the pressure and reduced anxiety, enabling the young people to be able to engage in some form of communication over time, as indicated in the literature ([14]; [30]).

### 4.4. Communication and Collaboration

Findings highlight the role of communication and collaboration as a key aspect of the school experience for CYP with SM. This includes implications across the individual and systems levels. Two of the recruited participants had co-occurring diagnoses of ASD, and as may be expected, they reported subsequent difficulties in relation to the social aspects of communication. In contrast to the current [25] ([25]) information, recent literature suggests a potential relationship between SM and ASD ([38]). These case studies further support the potential relationship, evidenced by the additional communication challenges experienced by Olivia and Catherine. Acknowledging this complex interplay supports educators to better understand the YP in a more holistic way, therefore informing the appropriate support strategies for the YP ([5]).

Research from [44] ([44]) highlighted the ‘tripartite role’ schools have in identifying and supporting children with SM. As part of this support, they acknowledged the need for collaboration within school, with parents and external professionals to establish informed, individualized support for children. This sentiment was echoed within the findings from this study, alongside the need for collaboration with the CYP themselves as they enter secondary school and become more independent and autonomous.

### 4.5. Individual Experience and Support

Another important concept that should inform support is the individual experiences of the CYP. [43] ([43]) gathered the views of CYP with SM using personal construct psychology and highlighted that CYP’s experiences reflected a combination of external school factors and internal factors that contributed to their success and difficulties within school. Similarly, within the findings of this study, CYP shared their experiences of a range of factors associated with the wider school experience, such as the role of staff, peers, the environment, and curriculum demands. CYP also shared a number of internal factors that appear to contribute to their experiences, such as anxiety levels, how they construe situations/events, and personal competencies with communication skills. A key finding was that Olivia faced more challenges with performative subjects, likely linked to comorbid Social Anxiety Disorder (SAD). SAD involves fear of social or performance situations, which can intensify difficulties for CYP with SM, especially in high-pressure or evaluative settings ([2]; [22]). Recognizing the overlap between SM and SAD is important for tailoring support, combining communication strategies with anxiety management to better meet individual needs in these contexts. It was only when there was a clear understanding of both these wider and intrinsic factors that support was helpful for the young people within school. Using this understanding, appropriate support within school appears to embody the principle of ‘the pendulum of challenge’, largely based on Vygotsky’s ‘Zone of proximal development’ ([42]). This enables educators to implement learning experiences and support that is tailored to the needs and abilities of the CYP, by pushing CYP just beyond their comfort zone into the zone of proximal development where they are able to ‘take risks’ and build confidence in school. Again, these findings further reinforce the key principle acknowledged by [5] ([5]) that support needs to be informed by a holistic understanding of the CYP, informed by potential environmental, intrinsic, and psychological factors.

### 4.6. Limitations and Future Research

Difficulties recruiting participants in this study suggest that many CYP with SM may currently be under-identified and, as a result, are not receiving appropriate support within school. Previous research and the findings within this paper highlight that this is most likely due to a misunderstanding of their ‘shy’ and ‘quiet’ presentation, leading to a lack of concern being raised by staff and/or parents ([19]). Several settings contacted during the project recruitment reported having no CYP with SM within their settings, despite their cohorts numbering around 1000–1500 pupils. Current prevalence rates would suggest there are at least 2–3 CYP within those settings with SM receiving no support ([37]). In line with the findings from this study, further work and research are needed to establish the most effective way to increase knowledge and understanding across education to support these CYP.

Despite gathering perspectives that are largely otherwise unrepresented, it is important to acknowledge that this study details exploratory case studies, resulting in highly contextualized data from each setting. The researcher recognizes that the bespoke nature of the data collection process meant that data were generated in an individualized way and were dependent upon the researcher–participant relationship, making the methods used less easily replicated. In addition, while research acknowledges the co-morbidity between SM and ASD, two of the three reported case studies also had diagnoses of ASD, further adding to the contextualized nature of the findings.

Similarly, due to these difficulties, it is important to acknowledge that the participant sample was largely made up of CYP and families that are receiving some support in school, and it appears to be parents who have been able to develop their knowledge of SM and the wider school system. Further research is needed to build upon the findings of this paper and explore a wider range of experiences, support packages, and home-school relationships in relation to SM.

## 5. Conclusions

School experience for CYP with SM is individual and relates to key internal, external, and wider themes. Understandably, holistic support that works to consider a wider understanding of the CYP, informed by potential environmental, intrinsic, and psychological factors, makes the school experience for these CYP more positive and generative. To be able to provide the most appropriate and effective package of support, all staff within a school need a basic understanding of SM. Embedding this awareness within school policy and teacher education is essential to ensure early identification and consistent, informed support across the whole school community. Systematic training and clear policies can foster inclusive environments that better meet the needs of all CYP with SM. This guides the development of trusted relationships through accessible communication methods. As relationships are established, CYP’s views can inform support further. Collaboration between home, school, and external agencies needs to be an embedded part of the approach. This ongoing collaboration among stakeholders ensures that support for CYP remains individualized, appropriate, and consistent.

## Figures and Tables

**Figure 1 behavsci-15-00947-f001:**
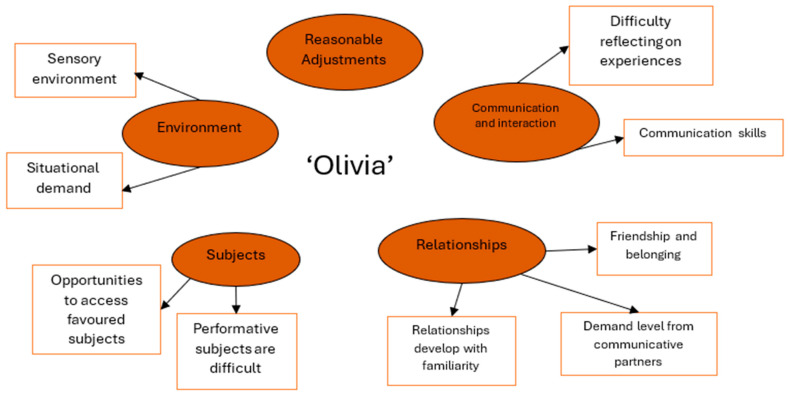
Thematic map of Olivia’s reported school experiences.

**Figure 2 behavsci-15-00947-f002:**
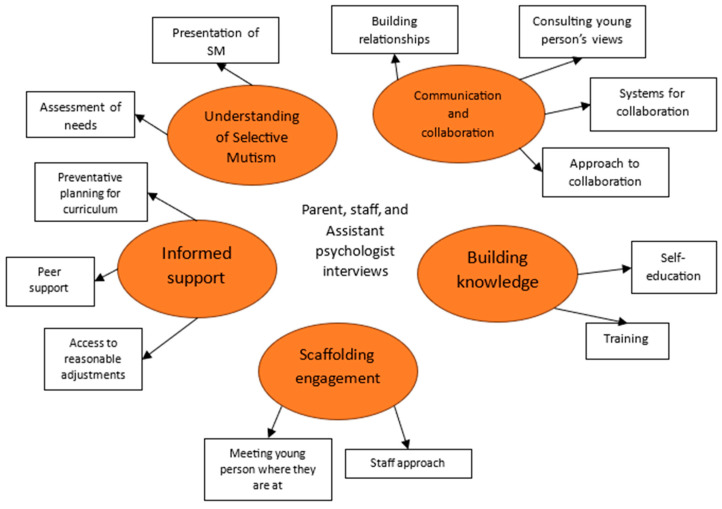
Thematic map of parent, staff, and assistant psychologist interviews.

**Figure 3 behavsci-15-00947-f003:**
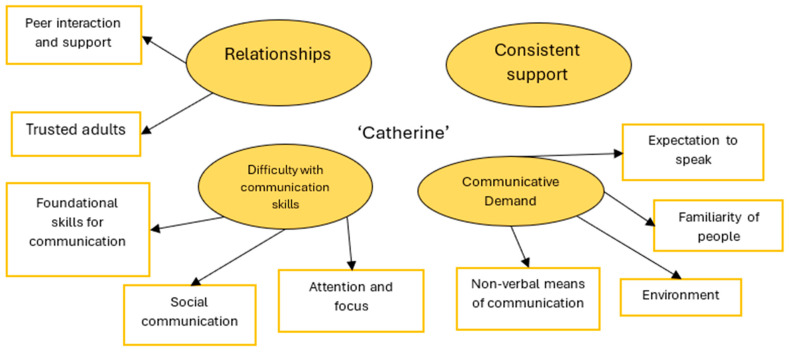
Thematic map of Catherine’s reported school experiences.

**Figure 4 behavsci-15-00947-f004:**
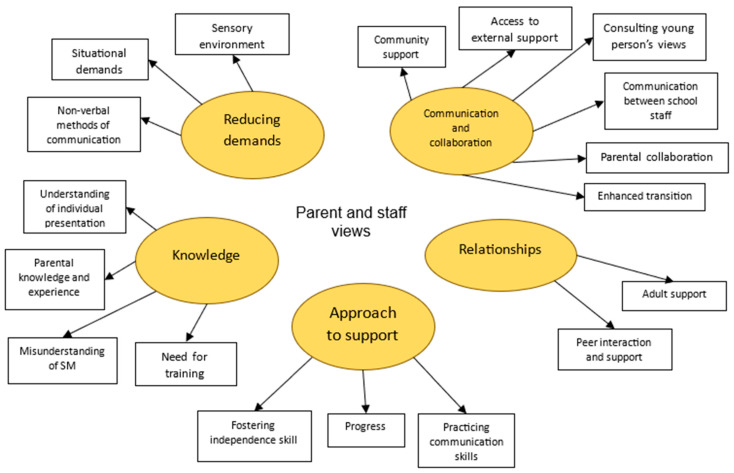
Thematic map of parent and staff interviews.

**Figure 5 behavsci-15-00947-f005:**
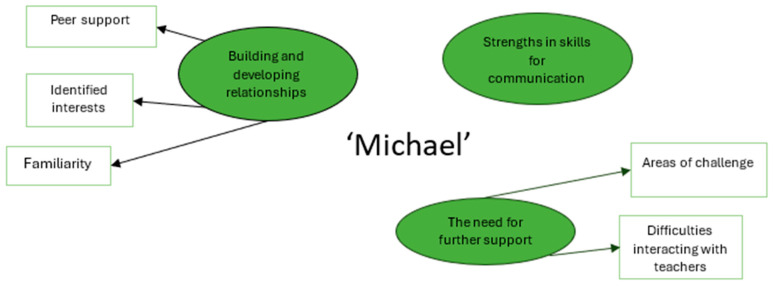
Thematic map of Michael’s reported school experiences.

**Figure 6 behavsci-15-00947-f006:**
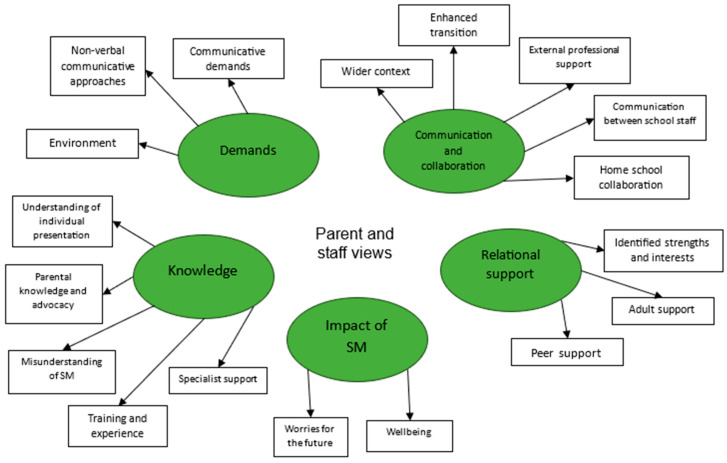
Thematic map of parent and staff interview responses.

**Table 1 behavsci-15-00947-t001:** Research Participants.

Case Study	Young Person Characteristics	School Characteristics	Stakeholders Recruited
Case study 1, Pseudonym Olivia	Female 13 years old Diagnosed with low-profile SM Co-occurring diagnoses of ASD and SAD (social anxiety disorder) No education, health and care plan in place in school	Elective grammar school Smaller than average number of pupils Average to higher levels of SEN in school	Special educational needs coordinator Parents (mother and father) Key worker Assistant Psychologist
Case study 2, Pseudonym Catherine Case study 3, Pseudonym Michael	Female 12 years old Diagnosed with SM Co-occurring diagnoses of ASD, anxiety, and medical needs Education, health and care plan in place Male 12 years old Diagnosed with selective mutism Education, health and care plan in place	School within a lower socioeconomic area Smaller than average number of pupils Significantly higher than average levels of SEN Rural area Average number of pupils Lower than average level of SEN in school	Special educational needs coordinator Mother Form tutor Special educational needs coordinator Mother Key worker

## Data Availability

The data sets presented in this article are not readily available due to ethical restrictions.

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
