# Peer review of "Exploring the Experiences and Current Support of Children and Young People with Selective Mutism Within Mainstream Secondary Schools"

_behavsci, 2025, doi:10.3390/bs15070947_

Round 1
Reviewer 1 Report
Comments and Suggestions for Authors
Manuscript title: Exploring the experiences and current support of children and young people with selective mutism within mainstream secondary schools.
Thank you for the opportunity to review this important research effort.
This manuscript aims to address an issue of increasing importance in mental health research: incorporating the lived and living experience of children and young people with mental health challenges (as well as the perspectives of their parents/carers, educators, and clinicians) into the empirical evidence base. The authors identified a gap in understanding CYP about selective mutism, and the study makes an original contribution to this knowledge gap.
The chosen methodology is appropriate to test the hypotheses, and the methods are presented clearly. This study is of interest to the readers, and there is overall merit in publishing this work.
General comments (3)
The article does not make reference to terminology typically used to describe this type of qualitative investigation (i.e., lived (past-tense) and living (present-tense) experience), or that it used it to inform the design or methods employed in the study to understand the meaning of the experience from the participant’s perspective. This study did both. (E.g., see World Health Organisation, 2002 / Sartor, 2023).
The discussion section addresses the cross-case analysis and the potential relationship between SM and ASD to explain the additional communication challenges experienced by Olivia and Catherine. However, it does not address the finding that ‘Olivia’ had more difficulties with performative subjects and the potential role of comorbid Social Anxiety disorder. I believe this is an important point that should be discussed.
I do not recommend self-referral of ‘the researcher’ throughout the method and result sections. Perhaps consider using ‘therapist’ or ‘clinician’ or first/second author.
In addition to one major concern, specific issues are listed in the attached document.

Major concern (1)
I believe this is an important piece of research, but it is poorly written. The manuscript would be greatly improved by a review of the grammar and sentence structures used within the manuscript. Currently, this important research is not presented as a finessed, academic-standard article.
Author Response
Please see attached document and revised manuscript. Thanks.

Reviewer 2 Report
Comments and Suggestions for Authors
The present study represents a valuable contribution to the literature on selective mutism (SM), especially in the underexplored context of secondary education. The adoption of a triangulated multi-case approach, which directly includes children's voices, is a notable methodological strength. Epistemological clarity, ethical sensitivity, and careful attention to context make this manuscript an example of high quality applied research.
Suggested for improvement:
Strengthen the comparative discussion between cases, pointing out patterns and divergences more clearly.
Briefly expand the applicability of the findings beyond the cases analysed, considering recommendations at the level of school policy or teacher education.
Congratulations on a solid and ethically grounded paper.
The manuscript is written in clear and generally well-structured academic English. Terminology is appropriate to the field, and key concepts are articulated with precision. Minor typographical and syntactic inconsistencies are present (e.g., inconsistent use of punctuation around citations, occasional word repetition), but these do not significantly affect readability or comprehension. A light language edit is recommended to further enhance fluency and clarity, particularly in the discussion section, where a more concise and focused exposition would strengthen the narrative flow.
Overall, the quality of the English is strong and does not hinder the delivery of the research findings.
Author Response

(The authors gave the same response as above.)
